# Shared Space Transfer Learning
# for analyzing multi-site fMRI data

**Muhammad Yousefnezhad**[1,2,3,*] **Alessandro Selvitella**[1,4]**, Daoqiang Zhang**[2]**,**
**Andrew J. Greenshaw**[1]**, Russell Greiner**[1,3]
[1]University of Alberta, Canada
[2]Nanjing University of Aeronautics and Astronautics, China
[3]Alberta Machine Intelligence Institute (Amii), Canada
[4]Purdue University Fort Wayne, United States
myousefnezhad@ualberta.ca, aselvite@pfw.edu, dqzhang@nuaa.edu.cn,
{andy.greenshaw, rgreiner}@ualberta.ca

## Abstract

Multi-voxel pattern analysis (MVPA) learns predictive models from task-based
functional magnetic resonance imaging (fMRI) data, for distinguishing when
subjects are performing different cognitive tasks — *e.g.*, watching movies or making
decisions. MVPA works best with a well-designed feature set and an adequate
sample size. However, most fMRI datasets are noisy, high-dimensional, expensive
to collect, and with *small sample sizes*. Further, training a robust, generalized
predictive model that can analyze homogeneous cognitive tasks provided by *multi-
site* fMRI datasets has additional challenges. This paper proposes the *Shared
Space Transfer Learning (SSTL)* as a novel transfer learning (TL) approach that
can functionally align homogeneous multi-site fMRI datasets, and so improve
the prediction performance in every site. SSTL first extracts a set of common
features for all subjects in each site. It then uses TL to map these site-specific
features to a site-*independent* shared space in order to improve the performance
of the MVPA. SSTL uses a scalable optimization procedure that works effectively
for high-dimensional fMRI datasets. The optimization procedure extracts the
common features for each site by using a single-iteration algorithm and maps
these site-specific common features to the site-independent shared space. We
evaluate the effectiveness of the proposed method for transferring between various
cognitive tasks. Our comprehensive experiments validate that SSTL achieves
superior performance to other state-of-the-art analysis techniques.

## 1 Introduction

The task-based functional magnetic resonance imaging (fMRI) is one of the prevalent tools in
neuroscience to analyze how human brains work [1–5]. It can be used to visualize the neural activities
when subjects are performing cognitive tasks — such as watching photos or making decisions [1].
Since brain images are high-dimensional and noisy, most of the recent neuroimage studies utilize
machine learning approaches such as classification techniques for analyzing fMRI datasets [1, 2].
Multi-voxel pattern analysis (MVPA) learns a classification model based on a set of fMRI responses
(with labels), which can be used to predict the cognitive tasks performed by a novel subject, who was
not part of the training phase [1]. An accurate MVPA model needs a well-designed feature space
and sufficient number of training instances [1–3]. However, most fMRI datasets include a limited set
of samples because collecting neuroimage data is an expensive procedure and needs a wide range

---

of agreements [2, 3]. As an alternative, increasing the number of publicly available fMRI datasets motivate the idea of combining multi-site *homogeneous* cognitive tasks (*i.e.*, all performing the same set of fMRI tasks) in order to increase the sample size, which we hope will boost the accuracy of the predictive models [2]. The best examples are the (U.S.) National Institute of Mental Health (NIMH)[2] [2] and Open NEURO[3] [7] projects that share thousands of fMRI scans with various types of cognitive tasks [2, 3].

It is challenging to train a generalized classification model from multi-site fMRI datasets, all involving the same set of homogeneous cognitive tasks [2, 3, 6]. There are two significant issues, *viz.*, differences in brain connectomes, and batch effects [2, 3]. As every human brain has a different connectome, each person will have a different neural response for the same stimulus [1]. Recent studies suggested applying functional alignment as an extra processing step before generating a prediction model for fMRI analysis [1, 3–5]. This *functional alignment* process extracts a set of common features from multi-subject fMRI data, which can be used to boost the prediction rate [3–5]. However, functional alignment techniques need temporal alignment — *i.e.*, the $i\text{-}th$ time point for all subjects must involve the same type of cognitive task [4, 5]. Although applying temporal alignment to a single-site fMRI data is a relatively straightforward process, this approach cannot be directly used for any multi-site datasets with different schemes of experimental designs [3, 4]. As another issue, *batch effects* [6] refers to a set of external elements that may affect the distribution of collected fMRI data in each site — *e.g.*, the environment noise, standards that are used by vendors of fMRI machines, etc. To deal with these issues, recent studies [2, 3, 8–10] show that *transfer learning (TL)* can significantly improve the quality of classification models for a multi-site fMRI analysis by leveraging the existing domain knowledge of the homogeneous cognitive tasks.

As the primary contribution of this paper, we propose *Shared Space Transfer Learning (SSTL)* as a novel TL approach that can generate a robust, generalized, accurate classification model from multi-site fMRI datasets, which can then be used effectively over each of these sites. SSTL learns a shared feature space by using a hierarchical two-step procedure. It first extracts a set of common features for all subjects in each site and then uses TL to map these site-specific features to a site-*independent* shared space. Further, SSTL uses a scalable optimization algorithm that works effectively for high-dimensional fMRI datasets with a large number of subjects. The optimization procedure extracts the common features for each site by using a single iteration multi-view approach and then maps these site common features to the site-independent shared space.

The rest of this paper is organized as follows: Section 2 briefly introduces some related works. Section 3 presents our proposed method. Section 4 reports the empirical studies, and finally, Section 5 presents the conclusion and points out some future works.

## 2   Related Works

Transfer learning (TL) has a wide range of applications in machine learning — *e.g.*, computer vision, or neural language processing [2, 3, 8–10]. However, most of TL techniques cannot be directly used for fMRI analysis [2]. There are several issues [2, 3]. First, fMRI signals (voxel values) have different properties in comparison with other types of data — such as natural images or texts [2]. In particular, the brain signals are highly-correlated with a low rate of the signal to noise ratio (SNR) that relies heavily on derived properties [4]. Moreover, each person has a different neural response for each individual stimulus because different brains have different connectomes [1, 2, 5]. Recent (single site) studies show that the neural responses of all subjects (in that site) can be considered as the noisy rotations of a common template [1, 3–5].

We use *homogeneous TL approaches* for task-based fMRI analysis, where the feature and label space in all sites are the same domain [2, 3]. These techniques minimize data distribution mismatch across all sites [2] — *i.e.*, mapping features of all sites to a shared space [3, 11, 12], or jointly learning a classification model and shifting all sites distributions that aim for a better accuracy rate [2, 13, 14]. Some TL techniques use a nonlinear transformation to fix the distribution mismatch — such as manifold embedded distribution alignment (MEDA) [14], AlexNet [15], and autoencoder [16]. Yan *et al.* recently developed the maximum independence domain adaptation (MIDA) [17] that uses the Hilbert-Schmidt Independence Criterion (HSIC) [18] to learn a set of common features across all

sites by minimizing statistical dependence on auxiliary domain side information [2, 17]. Inspired by the MIDA approach, Zhou *et al.* also proposed the Side Information Dependence Regularization (SIDeR) framework as a homogeneous TL designed for task-based fMRI analysis [2]. SIDeR uses HSIC and maximum mean discrepancy (MMD) to train a multi-site TL model that can simultaneously minimize the prediction risk and the mismatch on each site domain (common) information [2].

Multi-site fMRI analysis approaches can be seen in two ways — *viz.*, single-view methods [2, 6, 8–19], and multi-view techniques [3, 5]. As mentioned before, the same stimulus may imply distinctive neural responses because each brain has neurologically different connectome from other brains [1, 4, 5]. The single-view approaches do not accommodate these neurological differences between subjects of an individual site and consider all of them as a single distribution that must match with other sites distribution [2, 6, 8, 20]. Alternatively, the multi-view methods consider the neural activities belonging to each subject as a unique view and then learns a set of (site-specific) common features across all subjects [3–5]. Shared Response Model (SRM) [5] and Hyperalignment [1, 4] are the best examples of multi-view approaches that can align neural responses, but they work most effectively on an individual site. Recently, some studies showed that these techniques also could be used for transferring the cognitive tasks between multi-site fMRI datasets [2]. Based on these multi-view methods, Zhang *et al.* developed multi-dataset dictionary learning (MDDL) and multi-dataset multi-subject (MDMS) as two matrix factorization approaches that can learn accurate models for multi-site fMRI analysis [2]. MDDL uses a multi-view dictionary learning, and MDMS uses the probabilistic SRM [5] approach to generate the shared space [2]. Even though MDDL and MDMS can boost the performance accuracy, they are limited to transfer the cognitive tasks between multi-site fMRI datasets, as they require that some subjects appear in each pair of sites [3].

## 3 The Proposed Method

This section introduces the proposed *Shared Space Transfer Learning (SSTL)* as a novel TL approach that can improve the performance of the MVPA on homogeneous multi-site fMRI datasets. SSTL learns a TL model by using a hierarchical two-step procedure: It first extracts a set of *site-specific common features* for all subjects in each site and then transfers these common features to a *site-independent, global, shared space*. Unlike earlier models [3], SSTL does not require that some subjects appear in each pair of sites.

We let $D$ be the number of sites, $S_d$ be the number of subjects in $d$-*th* site, $T_d$ be the number of time-points in units of Time of Repetitions (TRs) for each subjects in $d$-*th* site, and $V$ be the number of voxels (which we view as a 1D vector, even though it corresponds to a 3D volume). The preprocessed brain image (neural responses) for $s$-*th* subject in $d$-*th* site is defined as $\mathbf{X}^{(d,s)} \in \mathbb{R}^{T_d \times V} = \left\{ x_{tv}^{(d,s)} \mid t = 1 \ldots T_d, v = 1 \ldots V \right\}$, $s = 1 \ldots S_d, d = 1 \ldots D$. In this paper, we make three assumptions. First, we assume that each column of the neural activities are standardized during preprocessing — *i.e.*, $\mathbf{X}^{(d,s)} \sim \mathcal{N}(0, 1)$, $s = 1 \ldots S_d$, $d = 1 \ldots D$. Second, temporal alignment is applied during preprocessing to neural responses of each site separately [1, 3–5]. Third, the $v$-*th* column in $\mathbf{X}^{(d,s)}$ denotes the anatomically aligned voxel that is located in the same locus for fMRI images in all sites [1–6]. For instance, we can register fMRI images of all sites to Montreal neurological institute (MNI) standard space and then apply the same mask to extract voxel values in the region of interest (ROI) or even use the standard whole-brain images.

### 3.1 Extracting site-specific common features

In this section, we develop an unsupervised multi-view method that can extract site-specific common features from every site separately. Let $k$ be the number of features in the common feature space. We calculate a mapping matrix $\mathbf{R}^{(d,s)} \in \mathbb{R}^{V \times k}$, $k \leq V$ to transform each subject neural responses to the common feature space $\mathbf{G}^{(d,S_d)} \in \mathbb{R}^{T_d \times k} = \left\{ g_{tv}^{(d,S_d)} \mid t = 1 \ldots T_d, v = 1 \ldots k \right\}$. We use following objective function to extract the mapping matrices and the common feature space for $d$-*th* site, where

$\mathbf{I}_k \in \{0,1\}^{k \times k}$ is the identity matrix of size $k$:

$$\mathcal{J}_C^{(d)}\left([\mathbf{X}^{(d,s)}]_{s=1\dots S_d}\right) = \underset{\mathbf{R}^{(d,s)}, \mathbf{G}^{(d,S_d)}}{\arg\min} \sum_{s=1}^{S_d} \left\| \mathbf{G}^{(d,S_d)} - \mathbf{X}^{(d,s)}\mathbf{R}^{(d,s)} \right\|_F^2,$$

$$\text{subject to} \quad \left(\mathbf{G}^{(d,S_d)}\right)^\top \mathbf{G}^{(d,S_d)} = \mathbf{I}_k. \tag{1}$$

We first propose regularized projection matrices and then use these matrices to estimate an optimal result for (1). We let $\epsilon$ be a regularization term, and $\mathbf{X}^{(d,s)} = \mathbf{U}^{(d,s)}\mathbf{\Sigma}^{(d,s)}\left(\mathbf{V}^{(d,s)}\right)^\top$ be the *rank-k* singular value decomposition (SVD) [23] of the neural responses. The regularized projection matrix belonging to the *s-th* subject in the *d-th* site is denoted by: [4, 21–23]

$$\mathbf{P}^{(d,s)} = \mathbf{X}^{(d,s)}\left(\mathbf{X}^{(d,s)}\left(\mathbf{X}^{(d,s)}\right)^\top + \epsilon\mathbf{I}_{T_d}\right)^{-1}\left(\mathbf{X}^{(d,s)}\right)^\top = \mathbf{U}^{(d,s)}\mathbf{\Phi}^{(d,s)}\left(\mathbf{U}^{(d,s)}\mathbf{\Phi}^{(d,s)}\right)^\top, \tag{2}$$

$$\mathbf{\Phi}^{(d,s)}\left(\mathbf{\Phi}^{(d,s)}\right)^\top = \mathbf{\Sigma}^{(d,s)}\left(\mathbf{\Sigma}^{(d,s)}\left(\mathbf{\Sigma}^{(d,s)}\right)^\top + \epsilon\mathbf{I}_{T_d}\right)^{-1}\left(\mathbf{\Sigma}^{(d,s)}\right)^\top. \tag{3}$$

**Lemma 1.** *Let* $\mathbf{R}^{(d,s)} = \left(\mathbf{X}^{(d,s)}\left(\mathbf{X}^{(d,s)}\right)^\top + \epsilon\mathbf{I}_{T_d}\right)^{-1}\left(\mathbf{X}^{(d,s)}\right)^\top \mathbf{G}^{(d,S_d)}, s = 1\dots S_d$ *be the transformation matrices for d-th site. Then, a regularized version of* (1) *can be written based on the common space* $\mathbf{G}^{(d,S_d)}$ *and the projection matrix* $\mathbf{P}^{(d,s)}$:

$$\widetilde{\mathcal{J}}_C^{(d)}\left([\mathbf{X}^{(d,s)}]_{s=1\dots S_d}\right) = \underset{\mathbf{G}^{(d,S_d)}}{\arg\max}\left(\text{tr}\left(\left(\mathbf{G}^{(d,S_d)}\right)^\top \sum_{s=1}^{S_d} \mathbf{P}^{(d,s)}\mathbf{G}^{(d,S_d)}\right)\right), \tag{4}$$

*Proof.* Please refer to supplemental material.

**Remark 1.** *We can consider no regularization term ($\epsilon = 0$) for calculating the projection matrices ($\mathbf{P}^{(d,s)}$), which implies $\widetilde{\mathcal{J}}_C^{(d)}\left([\mathbf{X}^{(d,s)}]_{s=1\dots S_d}\right) = \mathcal{J}_C^{(d)}\left([\mathbf{X}^{(d,s)}]_{s=1\dots S_d}\right)$. However, using no regularization may lead to an ill-posed analysis procedure. Since $\mathbf{X}^{(d,s)} \sim \mathcal{N}(0,1)$, the scatter matrices $\mathbf{X}^{(d,s)}\left(\mathbf{X}^{(d,s)}\right)^\top$ in (2) have properties of the Covariance matrices. Therefore, the scatter matrices are positive semidefinite and may be non-invertible [3–5, 21]. This is problematic especially when we select $k > \min(V, T_d)$ [4, 21]. Considering $\epsilon > 0$ as a positive regularization term can enable us to apply additional L2 regularization to (1) and skip the non-invertible issue — i.e., $\widetilde{\mathcal{J}}_C^{(d)}\left([\mathbf{X}^{(d,s)}]_{s=1\dots S_d}\right) \approx \mathcal{J}_C^{(d)}\left([\mathbf{X}^{(d,s)}]_{s=1\dots S_d}\right)$.*

Set $\mathbf{G}^{(d,0)} = \{0\}^{T_d \times k}$ to be the initial common space for *d-th* site, and let:

$$\mathbf{H}^{(d,s)} = \mathbf{P}^{(d,s)} - \mathbf{G}^{(d,s-1)}\left(\mathbf{G}^{(d,s-1)}\right)^\top \mathbf{P}^{(d,s)} \quad \text{for} \quad s = 1\dots S_d. \tag{5}$$

We calculate $\mathbf{H}^{(d,s)} = \mathbf{M}^{(d,s)}\mathbf{N}^{(d,s)}$ for $s = 1\dots S_d$ as QR decomposition of $\mathbf{H}^{(d,s)}$, where $\mathbf{M}^{(d,s)}\left(\mathbf{M}^{(d,s)}\right)^\top = \mathbf{I}_{T_d}$ and $\mathbf{N}^{(d,s)} \in \mathbb{R}^{T_d \times T_d}$ is an upper triangular matrix. Further, we let $\widetilde{\mathbf{\Sigma}}^{(d,s)}$ be diagonal matrices for all $s$, initialized as $\widetilde{\mathbf{\Sigma}}^{(d,0)} = \text{diag}(\{0\}^k)$. Then, we calculate:

$$\mathbf{A}^{(d,s)} = \left[\begin{array}{c|c} \widetilde{\mathbf{\Sigma}}^{(d,s-1)} & \left(\mathbf{G}^{(d,s-1)}\right)^\top \mathbf{P}^{(d,s)} \\ \hline \{0\}^{T_d \times k} & \mathbf{N}^{(d,s)} \end{array}\right] \quad \text{for} \quad s = 1\dots S_d. \tag{6}$$

Now, we let $\mathbf{A}^{(d,s)} = \widetilde{\mathbf{U}}^{(d,s)}\widetilde{\mathbf{\Sigma}}^{(d,s)}\left(\widetilde{\mathbf{V}}^{(d,s)}\right)^\top$ be the *rank-k* SVD decomposition [21, 22] for the generated matrices in (6) — with the left unitary matrix $\widetilde{\mathbf{U}}^{(d,s)} \in \mathbb{R}^{(T_d+k)\times k}$ and the diagonal matrix $\widetilde{\mathbf{\Sigma}}^{(d,s)} \in \mathbb{R}^k$. We then calculate following matrices:

$$\mathbf{B}^{(d,s)} = \left[\begin{array}{c|c} \mathbf{G}^{(d,s-1)} & \mathbf{M}^{(d,s)} \end{array}\right] \quad \text{for} \quad s = 1\dots S_d. \tag{7}$$

Finally, we have:

$$\mathbf{G}^{(d,s)} = \mathbf{B}^{(d,s)}\widetilde{\mathbf{U}}^{(d,s)} \quad \text{for} \quad s = 1\dots S_d. \tag{8}$$

Here, $\mathbf{G}^{(d,S_d)}$ for $d = 1\dots D$ are the site-specific common feature spaces that must be calculated for each site separately. Note that the main advantage of this approach is that there is an efficient way to update the common space, to incorporate a new subject: We only need to repeat the same procedure by using the current common space $\mathbf{G}^{(d,S_d)}$ and the data from the new subject $\mathbf{X}^{(d,S_d+1)}$.

## 3.2 Transferring site-specific common features to a global shared space

Our research aims to create a TL model for multi-site fMRI analysis by using site-specific common features, but not by directly transferring the raw neural responses [2] nor by finding a global shared space based on a set of subjects that are appeared in each pair of sites [3]. In this section, we partition our sites to a set of training sites and and a set of testing sites. We create a global shared space based on the common features of the training sites. We then use this global space to transfer neural responses in the testing site. We let $\widetilde{D}$ be the number of training sites, $\widetilde{T} = \sum_{d=1}^{\widetilde{D}} T_d$, and $\widehat{D}$ be the number of testing sites — i.e., $D = \widetilde{D} + \widehat{D}$. As the first step, we use the site-specific common spaces $\mathbf{G}^{(d,S_d)}$ for $d = 1 \dots \widetilde{D}$ to generate a global shared space. For simplicity, we denote a concatenated version of all common spaces in the training set as follows:

$$\mathbf{G} \in \mathbb{R}^{\widetilde{T} \times k} \quad = \quad \left\{ g_{tv} \mid t = 1 \dots \widetilde{T}, v = 1 \dots k \right\} \quad = \quad \begin{bmatrix} \mathbf{G}^{(1,S_1)} \\ \mathbf{G}^{(2,S_2)} \\ \vdots \\ \mathbf{G}^{(\widetilde{D},S_{\widetilde{D}})} \end{bmatrix}. \tag{9}$$

In this paper, we want to find a global shared space whose transformed common features have a minimum distribution mismatch. Let $\mathbf{g}_{t.} \in \mathbb{R}^k$ be the $t$-th row of matrix $\mathbf{G}$. Here, we want to find a pair of encoding/decoding transformation functions. The encoding transformation function $\mathbf{q}_{t.} = \mathcal{J}_1(\mathbf{g}_{t.}; \theta_1)$ maps the common features (from a specific site) into a global shared space — where $\mathbf{q}_{t.}$ is the $t$-th transformed common feature in the shared space. The decoding function $\bar{\mathbf{g}}_{t.} = \mathcal{J}_2(\mathbf{q}_{t.}; \theta_2)$ reconstructs the site-specific common features from the shared space. In general, we can use following objective function for finding these encoding/decoding transformations:

$$\mathcal{J}_G\left(\mathbf{G}\right) \quad = \quad \underset{\theta_1, \theta_2}{\arg\min} \sum_{t=1}^{\widetilde{T}} \left\| \mathbf{g}_{t.} - \bar{\mathbf{g}}_{t.} \right\|_F^2 + \Omega(\theta_1, \theta_2) \tag{10}$$

where $\Omega$ is a regularization function over the parameters $\theta_1, \theta_2$. There are several standard approaches in machine learning for solving (10). For instance, we can use regularized autoencoders for finding these transformations — where $\mathcal{J}_1, \mathcal{J}_1$ are the symmetric multilayer perceptron (MLP) [22]. However, complex models need a large number of instances to significantly boost the performance of analysis, and most of multi-site fMRI datasets do not have enough instances [2]. In this paper, we propose the following linear Karhunen–Loeve transformation (KLT) [24] for learning the global shared space:

$$\widetilde{\mathcal{J}}_G\left(\mathbf{G}\right) \quad = \quad \underset{\mathbf{W}}{\arg\min} \left\| \mathbf{G} - \mathbf{G}\mathbf{W}\mathbf{W}^\top \right\|_F^2, \tag{11}$$
$$\text{subject to} \quad \mathbf{W}^\top \mathbf{W} = \mathbf{I}_k.$$

where $\mathbf{W} \in \mathbb{R}^{k \times k}$ denotes a transformation matrix, $\mathbf{Q} = \mathcal{J}_1(\mathbf{G}; \mathbf{W}) = \mathbf{G}\mathbf{W}$, and $\bar{\mathbf{G}} = \mathcal{J}_2(\mathbf{Q}; \mathbf{W}^\top) = \mathbf{Q}\mathbf{W}^\top$. For finding the proposed shared space in (11), we first calculate zero-mean second moment of the matrix $\mathbf{G}$ as follows:

$$\mathbf{C} \quad = \quad \frac{1}{T-1} \left(\mathbf{G} - \mathbf{1}_T \mu^\top\right)^\top \left(\mathbf{G} - \mathbf{1}_T \mu^\top\right), \tag{12}$$

where $\mathbf{1}_T \in \{1\}^{T \times 1}$ denotes a column vector of all ones, row vector $\mu \in \mathbb{R}^{k \times 1}$ is the mean from each row of the matrix $\mathbf{G}$ — i.e., $v$-th element in vector $\mu$ is calculate by $\mu_v = \frac{1}{T} \sum_{t=1}^{T} g_{tv}$ for $v = 1 \dots k$. Finally, we can solve following eigendecomposition problem for finding the transformation matrix $\mathbf{W}$:

$$\mathbf{W}^\top \mathbf{C} \mathbf{W} = \mathbf{\Lambda} \tag{13}$$

where $\mathbf{\Lambda}$ and $\mathbf{W}$ are respectively the eigenvalues and eigenvectors of the matrix $\mathbf{C}$.

The SSTL learning procedure starts by generating the common feature space for each site separately — i.e., $\mathbf{G}^{(d,S_d)}$ for $d = 1 \dots D$. Then, it calculates the transformation matrix $\mathbf{W}$ by using the common feature in the training set, $\mathbf{G}^{(d,S_d)}$ for $d = 1 \dots \widetilde{D}$. Next, SSTL trains a classification model by using the transformed features in the training set — viz., $[\mathbf{X}^{(d,s)}\mathbf{R}^{(d,s)}\mathbf{W}]_{d=1\dots\widetilde{D}, s=1\dots S_d}$. Finally, we can evaluate the resulting multi-site TL model based on its accuracy on the transformed testing set $[\mathbf{X}^{(d,s)}\mathbf{R}^{(d,s)}\mathbf{W}]_{d=1\dots\widehat{D}, s=1\dots S_d}$. The Supplementary Material shows the SSTL learning procedure.

Table 1: The datasets.

| ID | Title (Open NEURO ID) | Type | $S_d$ | #1 | $T_d$ | #2 | #3 |
|---|---|---|---|---|---|---|---|
| A | Stop signal with spoken pseudo word naming (DS007) [25] | Decision | 20 | 4 | 149 | B, C | B, C, D |
| B | Stop signal with spoken letter naming (DS007) [25] | Decision | 20 | 4 | 112 | A, C | A, C, D |
| C | Stop signal with manual response (DS007) [25] | Decision | 20 | 4 | 211 | A, B | A, B, D |
| D | Conditional stop signal (DS008) [26] | Decision | 13 | 4 | 317 | | A, B, C |
| E | Simon task (DS101) (unpublished [2]) | Simon | 21 | 2 | 302 | | F |
| F | Flanker task (DS102) [27] | Flanker | 26 | 2 | 292 | | E |
| G | Integration of sweet taste – study 1 (DS229) [28] | Flavour | 15 | 6 | 580 | H | H |
| H | Integration of sweet taste – study 3 (DS231) [28] | Flavour | 9 | 6 | 650 | G | G |

$S_d$ is the number of subject; #1 is the number of stimulus categories; $T_d$ is the number of time points per subjects; #2 lists the other datasets that overlap with this dataset; #3 lists the other datasets whose neural responses can be transferred to this dataset.

## 4  Experiments

Table 1 lists the 8 datasets (A to H) used for our empirical studies. These datasets are provided by Open NEURO repository and are separately preprocessed by *easy fMRI*[4] and FSL 6.0.1[5] — *i.e.*, normalization, smoothing, anatomical alignment, temporal alignment. We also registered each scan to the MNI152 $T1$ standard space with voxel size 4mm, so total number of voxels is $V = 19742$. We shared all preprocessed datasets in the MATLAB format[6].

We compare SSTL with 6 different existing methods: raw neural responses in MNI space without using TL methods [3], the shared response model (SRM) [3, 5], the maximum independence domain adaptation (MIDA) [17], the Side Information Dependence Regularization (SIDeR) [2], the multi-dataset dictionary learning (MDDL) [3], and the multi-dataset multi-subject (MDMS) [3]. We used the implementations proposed in [3] for SRM, MDDL, and MDMS. Note those techniques are limited as they can only transfer cognitive tasks between multi-site fMRI datasets if some subjects appear in each pair of sites. Further, we used a PC with certain specifications[7] for generating the experimental studies. All algorithms for generating the experimental studies are shared as parts of our GUI-based toolbox called easy fMRI[4].

Like the previous studies [2–5, 22], we use $\nu$-support vector machine ($\nu$-SVM) [29] for classification analysis. Here, we only allow the neural responses belonging to a site to be used in either the training phase or testing phase, but not both. In the training phase, we use a one-subject-out strategy for each training site to generate the validation set — *i.e.*, all responses of a subject are considered as the validation set, and the other responses are used as the training set. We repeat the training phase until all subjects in a site have the chance to be the validation set. For instance, we want to learn a TL model from site A to predict all neural responses in site B (A → B). Here, we have $S_A$ subjects on site A. We learn $S_A$ independent models, where each of them is trained by the neural responses of all subjects in site A except the $s$-$th$ subject and tune by using neural responses of $s$-$th$ subject — $s \in \{1 \ldots S_A\}$. We evaluate each of these $S_A$ models on the neural responses of site B, then report the average of these $S_A$ evaluations as the performance of the "A → B" transference.

We used a scheme similar to the one proposed in [2–3] for evaluating all transfer learning approaches described in this paper. Our proposed SSTL first computes the unsupervised site-dependent $\mathbf{G}^{(d,S_d)}$, from the data, but not the labels, for all sites. Note this is similar to the procedures used in learning any classical functional alignment, such as SRM and HA. For classifying a subject in site $d$, SSTL then uses the labeled data from other $d-1$ sites to find the global shared space $\mathbf{W}$, then train the classifier — *n.b.*, using nothing from the $d$-$th$ site. Hence, SSTL never uses any labels from the $d$-$th$ site, when computing the labels for those $d$-$th$ site subjects. Like Westfall *et al.* [20], SSTL also used the standard learning procedure, *i.e.*, using a shuffled form of neural responses for training the classifier (not the temporally aligned version).

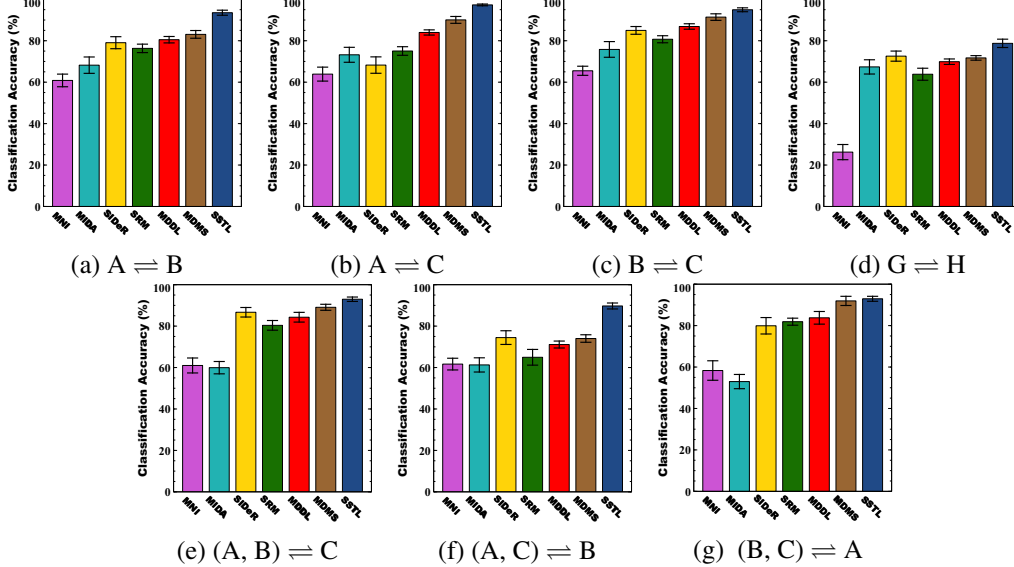

$$\text{(a) } A \rightleftharpoons B \qquad \text{(b) } A \rightleftharpoons C \qquad \text{(c) } B \rightleftharpoons C \qquad \text{(d) } G \rightleftharpoons H$$

$$\text{(e) } (A, B) \rightleftharpoons C \qquad \text{(f) } (A, C) \rightleftharpoons B \qquad \text{(g) } (B, C) \rightleftharpoons A$$

Figure 1: Multi-site classification analysis for datasets that overlap (*i.e.*, share some subjects). Error bars illustrate $\pm 1$ standard deviation.

We tune the hyper-parameters — regularization $\epsilon \in \{10^{-2}, 10^{-4}, 10^{-6}, 10^{-8}\}$, number of features $k$, maximum number of iterations $L$ — by using grid search based on the performance of the validation set. As mentioned before, SSTL just sets $L = 1$, but other TL techniques (such as SRM, MDDL, MSMD, etc.), we consider $L \in \{1, 2, ..., 50\}$ . For selecting the number of features $k$, we first let $k_1 = \min(V, T_d)$ for $d = 1 \ldots \widetilde{D}$ [4]. Then, we benchmark the performance of analysis by using $k = \alpha k_1$, where $\alpha = \{0.1, 0.5, 1, 1.1, 1.5, 2\}$. We report the performance evaluation of all one-way analysis (*e.g.*, A $\rightarrow$ B) in the Supplementary Material. For simplicity, we report the average of evaluations for each pair of these one-way analyses in the rest of this paper — *e.g.*, $acc($ A $\rightleftharpoons$ B $)$ denotes the average of evaluations for pairs $acc($ A $\rightarrow$ B $)$, and $acc($ B $\rightarrow$ A $)$.

### 4.1 Multi-site classification analysis for pairs of datasets that overlap

In this section, we report multi-site fMRI analysis by using datasets (*i.e.*, A, B, C, G, H) with some subjects that appear in each pair of sites. As mentioned before, SRM, MDDL, and MDMS require these pair-site subjects [3]. Here, we reported two levels of analysis — *viz.*, a peer to peer analysis, and the multi-site analysis. In a peer to peer analysis, we first learn a TL model from a single site to predict neural responses in another single site — *e.g.*, A $\rightleftharpoons$ B. Figures 1[a–d] illustrate the peer-to-peer analysis. We also benchmark the performance of TL models when there are more than two sites during an analysis — *e.g.*, (A, B) $\rightleftharpoons$ C. Figures 1[e–g] show the accuracies of these multi-site analyses, showing that the raw neural responses (in MNI space) perform poorly — perhaps because of the distribution mismatch in different sites. While single view approaches (MIDA and SIDeR) do perform better, they do not perform well on multi-site analysis. We see that the multi-view techniques in SRM, MDDL, MDMS, and SSTL enable them to generate TL models that are more accurate than the single view methods. Finally, SSTL provides most accurate TL models that lead to better performance, by (1) using a multi-view approach to generate the site-specific common features, then (2) using these common features (rather than the noisy raw neural responses) for transferring data to the global shared space. Note each of the 7 plots is comparing SSTL and $\chi_1$, for 6 different $\chi_1 \in \{$MNI, MIDA, SIDeR, SRM, MDDL, MDMS$\}$, for a total of $7 \times 6 = 42$ comparisons. A 2-sided t-test found $p <$0.05 in all $42$ cases.

### 4.2 Multi-site classification analysis for sets of datasets that do not overlap

In this section, we report more general multi-site fMRI analysis — where datasets have no subject that appears in each pair of sites. Therefore, we cannot use SRM, MDDL, and MDMS approaches to analyze these datasets [3]. Figure 2 shows the effect of different transfer learning approaches (*i.e.*, MIDA, SIDeR, and SSTL) on the performance of the multi-site fMRI analysis, with 2[a–d] showing

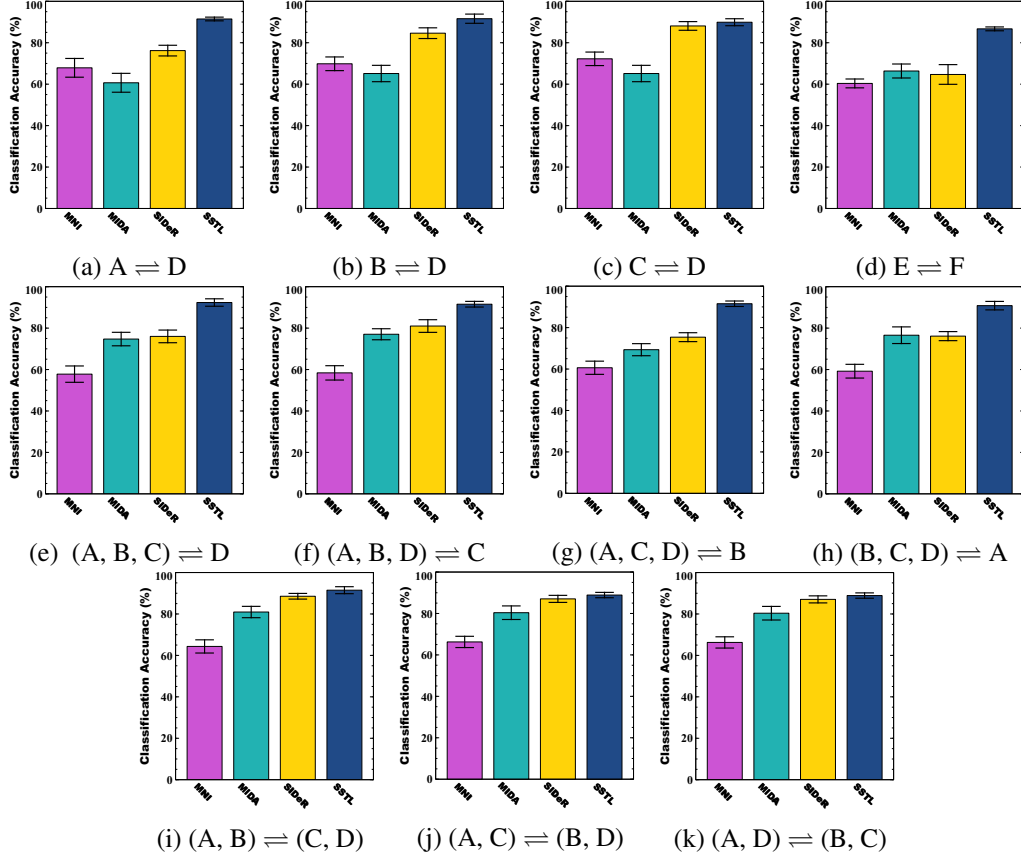

Figure 2: Multi-site classification analysis for datasets that have no overlap (*i.e.*, do not share any subjects). Error bars illustrate $\pm 1$ standard deviation.

the peer to peer analysis, and 2[e–k] illustrating the multi-site analysis. As shown in the previous studies [2, 3, 6, 9–16], these methods can improve the multi-site fMRI analysis results in comparison with pure classification models (in MNI space) without transfer learning. We see that our SSTL performs better than the single view approaches. These empirical results show that using site-specific common features for transferring multi-site fMRI datasets can boost the performance of the MPV analysis. Note each of the 11 plots is comparing SSTL and $\chi_2$, for 3 different $\chi_2 \in \{$MNI, MIDA, SIDeR$\}$, for a total of $11 \times 3 = 33$ comparisons. A 2-sided t-test found $p < 0.05$ in all 33 cases.

## 4.3 Runtime

This section analyzes the runtime of the various approaches on pairs of datasets (*i.e.*, pairs of A, B, C, G, and H) that overlap — *i.e.*, include some subjects in common. Here, all results are generated by using a PC with certain specifications[7]. Figure 3 shows the runtimes of the transfer learning techniques, scaled based on SSTL. We see that multi-view approaches (*i.e.*, MSMD, MDDL, SSA) had longer runtimes, probably because they must estimate the transformation matrices by using the iterative optimization techniques. The runtime of single-view approaches was better than the multi-view methods, perhaps because they process all instances in the (multi-site) training set together rather than analyzing neural responses belonging to each subject separately. This runtime is acceptable, given SSLT's superior classification performance (see Sections 4.1 and 4.2). Indeed, the main algorithmic difference between SSTL and the other techniques lies in the optimization procedure: While SSTL uses an efficient optimization procedure that calculates common space by applying a single-iteration method for each subject, the other transfer learning techniques used iterative optimization algorithms.

The Supplementary Material also reports the visualization of the transferred neural responses.

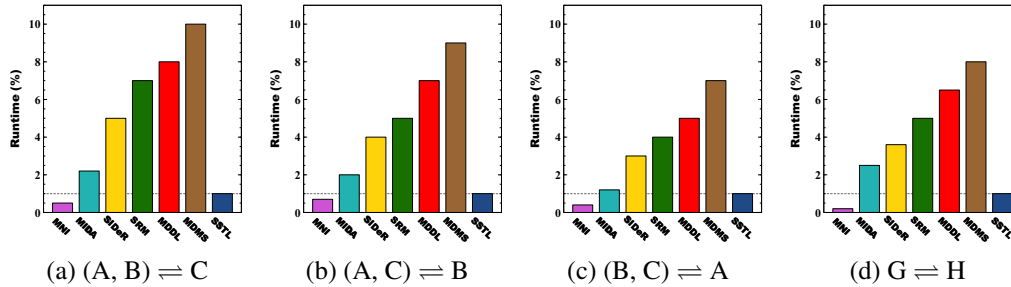

| (a) (A, B) ⇌ C | (b) (A, C) ⇌ B | (c) (B, C) ⇌ A | (d) G ⇌ H |

Figure 3: Runtime Analysis

# 5 Conclusion

In this paper, we propose the *Shared Space Transfer Learning (SSTL)* as a novel transfer learning (TL) technique that can be used for homogeneous multi-site fMRI analysis. SSTL first extracts a set of common features for all subjects in each site. It then uses TL to map these site-specific features to a global shared space, which improves the performance of the classification task. SSTL uses a scalable optimization procedure that can extract the common features for each site in a single pass through the subjects. It then maps these site-specific common features to the site-independent shared space. To the best of our knowledge, SSTL is the only multi-view transfer learning approach that can be used for analyzing multi-site fMRI datasets — which have no subjects that appear in each pair of sites. We evaluate the effectiveness of the proposed method for transferring between various cognitive tasks — such as decision making, flavor assessment, etc. Our comprehensive experiments confirmed that SSTL achieves superior performance to other state-of-the-art TL analysis methods. We anticipate that SSTL's multi-view technique for transfer learning will have strong practical applications in neuroscience — such as functional alignment of multi-site fMRI data, perhaps of movie stimuli.

**Acknowledgments**

This work was supported by the National Natural Science Foundation of China (Nos. 61876082, 61732006, 61861130366), the National Key R&D Program of China (Grant Nos. 2018YFC2001600, 2018YFC2001602, 2018ZX10201002), the Research Fund for International Young Scientists of China (NSFC Grant No. 62050410348), the Royal Society-Academy of Medical Sciences Newton Advanced Fellowship (No. NAF\R1\180371), the Natural Sciences and Engineering Research Council (NSERC) of Canada, and the Alberta Machine Intelligence Institute (Amii).

# Broader Impact

In this paper, we develop the Shared Space Transfer Learning (SSTL) as a novel transfer learning (TL) approach that can functionally align homogeneous multi-site fMRI datasets and so improve the prediction performance in every site. Although the proposed method is used to analyzed multi-site fMRI datasets, it can also be seen as a general-purpose machine learning method for any multi-view domain adaption applications. The proposed method is evaluated by using publicly-available fMRI datasets —- provided by Open NEURO. SSTL is an open-source technique and can also be used via our GUI-based toolbox called easy fMRI. We do not anticipate any negative consequences for this study. We believe that SSTL's multi-view technique for transfer learning will have strong practical applications — including, but not limited, neuroscience, computational psychiatry, human-brain interface, etc. In the future, we plan to utilize the proposed framework to analyze high-level cognitive processes such as movie stimuli.

## Footnotes

[2]Available at `https://data-archive.nimh.nih.gov/`

[3]Available at `https://openneuro.org/`

[4]https://easyfmri.learningbymachine.com/

[5]https://fsl.fmrib.ox.ac.uk/

[6]Available at https://easydata.learningbymachine.com/

[7] Main: Giga X399, CPU: AMD Ryzen Threadripper 2920X (24×3.5 GHz), RAM: 128GB, GPU: NVIDIA GeForce RTX 2080 SUPER (8GB memory), OS: Fedora 33, Python: 3.8.5, Pip: 20.2.3, Numpy: 1.19.2, Scipy: 1.5.2, Scikit-Learn: 0.23.2, MPI4py: 3.0.3, PyTorch: 1.6.0.

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
