[Supplementary Material 1]

# Shared Space Transfer Learning
# for analyzing multi-site fMRI data

### Supplemental Material

**Muhammad Yousefnezhad[1,2,3,*] Alessandro Selvitella[1,4], Daoqiang Zhang[2],**
**Andrew J. Greenshaw[1], Russell Greiner[1,3]**
[1]University of Alberta, Canada
[2]Nanjing University of Aeronautics and Astronautics, China
[3]Alberta Machine Intelligence Institute (Amii), Canada
[4]Purdue University Fort Wayne, United States
myousefnezhad@ualberta.ca, aselvite@pfw.edu, dqzhang@nuaa.edu.cn,
{andy.greenshaw, rgreiner}@ualberta.ca

## A  Proofs

**Lemma 1**
*We assume* $\mathbf{R}^{(d,s)} = \left(\mathbf{X}^{(d,s)}\left(\mathbf{X}^{(d,s)}\right)^\top + \epsilon \mathbf{I}_{T_d}\right)^{-1}\left(\mathbf{X}^{(d,s)}\right)^\top \mathbf{G}^{(d,S_d)}, s = 1 \ldots S_d$ *are optimum transformation matrices for* $d$-th *site. Then, a regularized version of (1) can be written based on the common space* $\mathbf{G}^{(d,S_d)}$ *and the projection matrix* $\mathbf{P}^{(d,s)}$:

$$\widetilde{\mathcal{J}}_C^{(d)}\left([\mathbf{X}^{(d,s)}]_{s=1\ldots S_d}\right) \quad = \quad \underset{\mathbf{G}^{(d,S_d)}}{\arg\max}\left(\mathrm{tr}\left(\left(\mathbf{G}^{(d,S_d)}\right)^\top \sum_{s=1}^{S_d} \mathbf{P}^{(d,s)}\mathbf{G}^{(d,S_d)}\right)\right).$$

*Proof.* In this paper, we use the same procedure proposed in [1–3]. We start from (1):

$$\mathcal{J}_C^{(d)}\left([\mathbf{X}^{(d,s)}]_{s=1\ldots S_d}\right) \quad = \quad \underset{\mathbf{R}^{(d,s)}, \mathbf{G}^{(d,S_d)}}{\arg\min} \sum_{s=1}^{S_d} \left\|\mathbf{G}^{(d,S_d)} - \mathbf{X}^{(d,s)}\mathbf{R}^{(d,s)}\right\|_F^2 \qquad (1)$$

By substituting $\mathbf{R}^{(d,s)} = \left(\mathbf{X}^{(d,s)}\left(\mathbf{X}^{(d,s)}\right)^\top + \epsilon \mathbf{I}_{T_d}\right)^{-1}\left(\mathbf{X}^{(d,s)}\right)^\top \mathbf{G}^{(d,S_d)}$, we have:

$$\mathcal{J}_C^{(d)}\left([\mathbf{X}^{(d,s)}]_{s=1\ldots S_d}\right) \quad \approx \quad \widetilde{\mathcal{J}}_C^{(d)}\left([\mathbf{X}^{(d,s)}]_{s=1\ldots S_d}\right)$$

$$= \quad \underset{\mathbf{G}^{(d,S_d)}}{\arg\min} \sum_{s=1}^{S_d} \left\|\mathbf{G}^{(d,S_d)} - \mathbf{X}^{(d,s)}\left(\mathbf{X}^{(d,s)}\left(\mathbf{X}^{(d,s)}\right)^\top + \epsilon \mathbf{I}_{T_d}\right)^{-1}\left(\mathbf{X}^{(d,s)}\right)^\top \mathbf{G}^{(d,S_d)}\right\|_F^2$$

---

$^*$Corresponding author.

**Algorithm 1** Shared Space Transfer Learning (SSTL)
___
**Input:**

   Training set $[\mathbf{X}^{(d,s)}]_{d=1...\widetilde{D},s=1...S_d}$,

   Training labels $[\mathbf{y}^{(d,s)}]_{d=1...\widetilde{D},s=1...S_d}$,

   Testing set $[\mathbf{X}^{(d,s)}]_{d=1...\widehat{D},s=1...S_d}$,

   Testing labels $[\mathbf{y}^{(d,s)}]_{d=1...\widehat{D},s=1...S_d}$,

   Regularized parameter $\epsilon$,

   Number of features $k$.

**Output:**

   Classification Model $\mathbf{\Pi}$,

   Site-specific common features $[\mathbf{G}^{(d,S_d)}]_{d=1...\widetilde{D}+\widehat{D}}$,

   Global shared space transformation $\mathbf{W}$,

   and the model evaluation (accuracy, precision, etc.).

**Method:**

<center># Common Phase — must run for each dataset separately</center>

01. $D = \widetilde{D} + \widehat{D}$
02. Initialize $\mathbf{G}^{(d,0)} = \{0\}^{T_d \times k}$ and $\widetilde{\mathbf{\Sigma}}^{(d,0)} = \mathrm{diag}(\{0\}^k)$ for $d = 1 \dots D$.
03. Generate $\mathbf{G}^{(d,S_d)}$ and $\mathbf{R}^{(d,s)}$ for $d = 1 \dots D$ and $s = 1 \dots S_d$ by using (1) to (8).

<center># Training Phase</center>

04. Concatenate $\mathbf{G} = [\mathbf{G}^{(d,S_d)}]_{d=1...\widetilde{D}}$ based on (9).

05. Calculate the second moment $\mathbf{C} = \frac{1}{T-1}\left(\mathbf{G} - \mathbf{1}_T \mu^\top\right)^\top \left(\mathbf{G} - \mathbf{1}_T \mu^\top\right)$ based on (12).

06. Calculate $\mathbf{W}$ as eigenvectors of $\mathbf{C}$.

07. Train a classification model $\mathbf{\Pi}\left([\mathbf{X}^{(d,s)}\mathbf{R}^{(d,s)}\mathbf{W}]_{d=1...\widetilde{D},s=1...S_d}, [\mathbf{y}^{(d,s)}]_{d=1...\widetilde{D},s=1...S_d}\right).$

<center># Testing Phase</center>

08. Predict based on model $[\widehat{\mathbf{p}}^{(d,s)}]_{d=1...\widehat{D},s=1...S_d} = \mathbf{\Pi}\left([\mathbf{X}^{(d,s)}\mathbf{R}^{(d,s)}\mathbf{W}]_{d=1...\widehat{D},s=1...S_d}\right).$

09. Evaluate accuracy of the model — *i.e.*, $[\widehat{\mathbf{p}}^{(d,s)}]_{d=1...\widehat{D},s=1...S_d}$ vs. $[\mathbf{y}^{(d,s)}]_{d=1...\widehat{D},s=1...S_d}.$
___

By considering the definition of the regularized projection matrix $\mathbf{P}^{(d,s)}$, we have:

$$\underset{\mathbf{G}^{(d,S_d)}}{\arg\min} \sum_{s=1}^{S_d} \left\|\mathbf{G}^{(d,S_d)} - \mathbf{P}^{(d,s)}\mathbf{G}^{(d,S_d)}\right\|_F^2 = \underset{\mathbf{G}^{(d,S_d)}}{\arg\min} \sum_{s=1}^{S_d} \left\|\left(\mathbf{I}_{T_d} - \mathbf{P}^{(d,s)}\right)\mathbf{G}^{(d,S_d)}\right\|_F^2$$

$$= \underset{\mathbf{G}^{(d,S_d)}}{\arg\min} \sum_{s=1}^{S_d} \mathrm{tr}\left(\left(\left(\mathbf{I}_{T_d} - \mathbf{P}^{(d,s)}\right)\mathbf{G}^{(d,S_d)}\right)^\top \left(\mathbf{I}_{T_d} - \mathbf{P}^{(d,s)}\right)\mathbf{G}^{(d,S_d)}\right)$$

$$= \underset{\mathbf{G}^{(d,S_d)}}{\arg\min} \sum_{s=1}^{S_d} \mathrm{tr}\left(\left(\mathbf{G}^{(d,S_d)}\right)^\top \left(\mathbf{I}_{T_d} - \mathbf{P}^{(d,s)}\right)^\top \left(\mathbf{I}_L - \mathbf{P}^{(d,s)}\right)\mathbf{G}^{(d,S_d)}\right)$$

$$= \underset{\mathbf{G}^{(d,S_d)}}{\arg\min} \sum_{s=1}^{S_d} \mathrm{tr}\left(\left(\mathbf{G}^{(d,S_d)}\right)^\top \left(\mathbf{I}_{T_d} - \mathbf{P}^{(d,s)}\right)^2 \mathbf{G}^{(d,S_d)}\right)$$

$$= \underset{\mathbf{G}^{(d,S_d)}}{\arg\min} \sum_{s=1}^{S_d} \mathrm{tr}\left(\left(\mathbf{G}^{(d,S_d)}\right)^\top \left(\mathbf{I}_{T_d}^2 + \left(\mathbf{P}^{(d,s)}\right)^2 - 2\mathbf{I}_{T_d}\mathbf{P}^{(d,s)}\right)\mathbf{G}^{(d,S_d)}\right)$$

Since $\mathbf{P}^{(d,s)}$ is idempotent $\left(\left(\mathbf{P}^{(d,s)}\right)^2 = \mathbf{P}^{(d,s)}\right)$ [1–3], we have:

$$= \underset{\mathbf{G}^{(d,S_d)}}{\arg\min} \sum_{s=1}^{S_d} \mathrm{tr}\left(\left(\mathbf{G}^{(d,S_d)}\right)^\top \left(\mathbf{I}_{T_d}^2 + \mathbf{P}^{(d,s)} - 2\mathbf{P}^{(d,s)}\right)\mathbf{G}^{(d,S_d)}\right)$$

$$= \underset{\mathbf{G}^{(d,S_d)}}{\arg\min} \sum_{i=s}^{S_d} \mathrm{tr}\left(\left(\mathbf{G}^{(d,S_d)}\right)^\top \left(\mathbf{I}_{T_d} - \mathbf{P}^{(d,s)}\right)\mathbf{G}^{(d,S_d)}\right)$$

<center>2</center>

$$= \underset{\mathbf{G}^{(d,S_d)}}{\arg\min} \left( \mathrm{tr}\left( \left(\mathbf{G}^{(d,S_d)}\right)^\top \left(\sum_{s=1}^{S_d} \mathbf{I}_{T_d} - \mathbf{P}^{(d,s)}\right) \mathbf{G}^{(d,S_d)} \right) \right)$$

$$\equiv \underset{\mathbf{G}^{(d,S_d)}}{\arg\max} \left( \mathrm{tr}\left( \left(\mathbf{G}^{(d,S_d)}\right)^\top \left(\sum_{s=1}^{S_d} \mathbf{P}^{(d,s)}\right) \mathbf{G}^{(d,S_d)} \right) \right) \quad \blacksquare$$

## B SSTL Algorithm

Algorithm 1 shows the Shared Space Transfer Learning (SSTL) learning procedure.

## C Notations

Table 1 and Table 2 respectively show all variables and functions that are used in our paper.

Table 1: Variables

| Variable or Function | Description |
|---|---|
| $\mathbb{R}$ | The set of real numbers. |
| $\mathbf{I}_k$ | The identity matrix in size $k$. |
| $\mathbf{1}_T$ | All-ones vector in size $T$. |
| $d$ | Index of the $d$-$th$ site index. |
| $D$ | Total of all sites in a multi-site fMRI analysis. |
| $\widetilde{D}$ | Number of training sites in a multi-site fMRI analysis. |
| $\widehat{D}$ | Number of testing sites in a multi-site fMRI analysis. |
| $s$ | Index of the $s$-$th$ subject. |
| $S_d$ | Total of all subjects in the $d$-$th$ site. |
| $v$ | Index of the $v$-$th$ voxel or feature. |
| $V$ | Total of all voxels in each dataset. |
| $k$ | Total of all extracted features. |
| $T_d$ | Total of all time points for the $s$-$th$ subject in the $d$-$th$ dataset. |
| $\epsilon$ | A regularization term. |
| $L$ | Maximum number of iterations (for other TL techniques) |
| $\widetilde{T} = \sum_{d=1}^{\widetilde{D}} T_d$ | Total of all time points in the training sites. |
| $\mathbf{X}^{(d,s)} \in \mathbb{R}^{T_d \times V}$ | Matrix of neural responses for $s$-$th$ subject in $d$-$th$ site. |
| $\mathbf{y}^{(d,s)} \in \mathbb{R}^{T_d}$ | Label vector for $s$-$th$ subject in $d$-$th$ site. |
| $x_{tv}^{(d,s)}$ | A neural response scalar for $t$-$th$ time point, $v$-$th$ voxel, $s$-$th$ subject, $d$-$th$ site. |
| $\mathbf{R}^{(d,s)} \in \mathbb{R}^{V \times k}$ | Site-specific mapping matrix for $s$-$th$ subject in $d$-$th$ site. |
| $\mathbf{G}^{(d,s)} \in \mathbb{R}^{T_d \times k}$ | Site-specific common feature for $s$-$th$ subject in $d$-$th$ site. |
| $\mathbf{G}^{(d,S_d)} \in \mathbb{R}^{T_d \times k}$ | Final site-specific common feature for $d$-$th$ site. |
| $g_{tv}^{(d,S_d)}$ | Final site-specific scalar for $t$-$th$ time point, $v$-$th$ voxel, $d$-$th$ site. |
| $\mathbf{G} \in \mathbb{R}^{\widetilde{T} \times k}$ | A concatenated version of all common features in the training set. |
| $\mathbf{g}_{t.}, \bar{\mathbf{g}}_{t.}$ | The common feature for $t$-$th$ time point, $v$-$th$ voxel, $d$-$th$ site. |
| $\mathbf{P}^{(d,s)}$ | Projection matrix for $s$-$th$ subject in $d$-$th$ site. |
| $\mathbf{\Phi}^{(d,s)}$ | A diagonal matrix is used for calculating projection for $s$-$th$ subject in $d$-$th$ site. |
| $\mathbf{H}^{(d,s)}, \mathbf{A}^{(d,s)}, \mathbf{B}^{(d,s)}$ | Matrices used for optimization procedure for $s$-$th$ subject in $d$-$th$ site. |
| $\mathbf{U}^{(d,s)}, \widetilde{\mathbf{U}}^{(d,s)}$ | Left unitary matrices generated respectively by SVD decomposition of $\mathbf{X}^{(d,s)}$ and $\mathbf{A}^{(d,s)}$ |
| $\mathbf{\Sigma}^{(d,s)}, \widetilde{\mathbf{\Sigma}}^{(d,s)}$ | Rectangular diagonal matrices generated respectively by SVD decomposition of $\mathbf{X}^{(d,s)}$ and $\mathbf{A}^{(d,s)}$ |
| $\mathbf{V}^{(d,s)}, \widetilde{\mathbf{V}}^{(d,s)}$ | Right unitary matrices generated respectively by SVD decomposition of $\mathbf{X}^{(d,s)}$ and $\mathbf{A}^{(d,s)}$ |
| $\mathbf{M}^{(d,s)}\mathbf{N}^{(d,s)}$ | QR decomposition of $\mathbf{H}^{(d,s)}$; $\mathbf{N}^{(d,s)} \in \mathbb{R}^{T_d \times T_d}$ is an upper triangular matrix. |
| $\mathbf{C}$ | Zero-mean second moment of the matrix $\mathbf{G}$ |
| $\mathbf{W} \in \mathbb{R}^{k \times k}$ | A linear transformation to a global shared space; Eigenvectors of $\mathbf{C}$ |
| $\mathbf{Q} \in \mathbb{R}^{\widetilde{T} \times k}$ | Mapped common features in the global shared space; $\mathbf{Q} = \mathbf{G}\mathbf{W}$ |
| $\mathbf{\Lambda}$ | Eigenvalues of $\mathbf{C}$. |
| $\mu \in \mathbb{R}^{k \times 1}$ | The mean vector from each row of the matrix $\mathbf{G}$. |

Table 2: Functions

| Variable or Function | Description |
|---|---|
| $\mathcal{J}_C^{(d)}$ | The general objective function for extracting site-specific common space for $d\text{-}th$ site. |
| $\tilde{\mathcal{J}}_C^{(d)}$ | The regularized function for extracting site-specific common space for $d\text{-}th$ site. |
| $\mathcal{J}_G$ | The general objective function for extracting global shared space. |
| $\tilde{\mathcal{J}}_G$ | The Karhunen–Loeve transformation (KLT) for extracting global shared space. |
| $\mathcal{N}(0,1)$ | Normalized distribution with zero mean and unit variance. |
| $diag()$ | Diagonal function convert a vector to a diagonal matrix. |
| $tr()$ | Trace function. |
| $\Omega()$ | Regularization function over hyper-parameters. |
| $\mathbf{\Pi}()$ | A classification model. |
| $\|.\|_F$ | The Frobenius norm. |

Table 3: Accuracy of classification analysis (mean±std). The best result for each task is in **bold**, and the second best is underlined.

| Datasets | MNI | MIDA | SIDeR | SRM | MDDL | MDMS | SSTL |
|---|---|---|---|---|---|---|---|
| A → B | 62.51±2.4 | 65.44±4.21 | 78.01±2.7 | 75.64±2.41 | 78.97±1.89 | 81.34±1.73 | **94.65±1.42** |
| B → A | 59.17±3.7 | 70.98±3.69 | 80.1±3.1 | 76.99±1.67 | 82.04±1.21 | 84.8±2.00 | **92.40±1.02** |
| A → C | 67.31±1.6 | 77.19±3.46 | 67.39±4.01 | 75.27±2.17 | 84.00±1.04 | 90.69±1.37 | **98.18±0.26** |
| C → A | 60.47±5.14 | 69.24±3.76 | 69.12±3.91 | 74.86±1.98 | 83.99±1.47 | 89.43±1.94 | **96.42±0.59** |
| B → C | 69.18±2.47 | 79.26±4.1 | 91.02±1.24 | 81.45±2.39 | 90.47±1.27 | 93.23±1.64 | **95.72±1.07** |
| C → B | 61.82±1.93 | 72.35±3.47 | 78.96±2.50 | 79.99±1.01 | 83.29±1.37 | 89.60±1.53 | **94.18±0.81** |
| H → G | 25.11±3.07 | 71.24±3.81 | 70.04±2.17 | 65.33±1.87 | 69.61±1.14 | 70.96±1.20 | **78.35±1.04** |
| G → H | 27.38±4.26 | 63.51±3.07 | 75.10±2.71 | 62.34±3.92 | 70.16±1.72 | 72.46±1.00 | **79.14±0.98** |
| (A, B) → C | 67.34±2.71 | 66.59±2.07 | 88.99±1.91 | 81.36±2.37 | 85.94±2.16 | 90.00±1.46 | **94.62±1.02** |
| C → (A, B) | 52.73±4.51 | 53.29±3.88 | 84.38±2.69 | 79.37±2.31 | 82.64±2.53 | 88.16±1.45 | **91.33±1.10** |
| (A, C) → B | 64.94±2.01 | 62.46±2.94 | 79.50±2.00 | 65.12±3.46 | 69.38±1.91 | 77.81±1.53 | **90.03±0.81** |
| B → (A, C) | 58.39±3.64 | 60.08±3.92 | 69.41±4.61 | 64.78±4.13 | 72.84±1.43 | 70.21±2.10 | **89.41±2.06** |
| (B, C) → A | 59.27±4.00 | 52.94±2.76 | 81.23±3.41 | 82.99±1.27 | 85.34±2.86 | 91.81±1.84 | **94.05±1.04** |
| A → (B, C) | 57.37±5.41 | 53.03±4.11 | 78.61±4.52 | 80.79±2.13 | 82.19±3.24 | **92.04±2.59** | 91.84±1.36 |
| A → D | 73.33±3.11 | 67.70±5.10 | 86.26±1.95 | *N/A* | *N/A* | *N/A* | **92.09±0.74** |
| D → A | 62.45±5.94 | 53.64±4.01 | 66.15±3.20 | *N/A* | *N/A* | *N/A* | **90.84±1.05** |
| B → D | 74.61±4.15 | 61.78±4.72 | **93.81±2.17** | *N/A* | *N/A* | *N/A* | 92.17±2.07 |
| D → B | 65.07±2.46 | 68.51±3.17 | 75.42±3.0 | *N/A* | *N/A* | *N/A* | **91.04±2.37** |
| C → D | 73.91±3.67 | 74.16±4.60 | 88.52±2.02 | *N/A* | *N/A* | *N/A* | **90.41±1.40** |
| D → C | 70.57±2.85 | 75.07±3.88 | 87.69±2.17 | *N/A* | *N/A* | *N/A* | **89.39±1.97** |
| E → F | 66.07±1.81 | 69.55±3.70 | 74.00±3.87 | *N/A* | *N/A* | *N/A* | **85.39±1.07** |
| F → E | 54.66±2.46 | 63.17±3.05 | 55.37±5.64 | *N/A* | *N/A* | *N/A* | **88.01±0.79** |
| (A, B, C) → D | 54.58±4.38 | 78.91±2.83 | 75.42±3.95 | *N/A* | *N/A* | *N/A* | **95.29±1.21** |
| D → (A, B, C) | 60.99±3.51 | 70.52±3.68 | 74.82±2.16 | *N/A* | *N/A* | *N/A* | **89.41±2.38** |
| (A, B, D) → C | 57.71±2.09 | 78.77±2.41 | 80.57±2.09 | *N/A* | *N/A* | *N/A* | **94.36±1.27** |
| C → (A, B, D) | 59.08±4.83 | 75.30±2.91 | 79.62±1.26 | *N/A* | *N/A* | *N/A* | **88.71±1.47** |
| (A, C, D) → B | 61.83±3.81 | 70.39±3.12 | 76.97±2.05 | *N/A* | *N/A* | *N/A* | **92.69±1.35** |
| B → (A, C, D) | 59.41±2.56 | 68.29±2.71 | 73.80±2.25 | *N/A* | *N/A* | *N/A* | **90.36±1.28** |
| (B, C, D) → A | 58.39±2.71 | 77.92±3.63 | 76.18±2.91 | *N/A* | *N/A* | *N/A* | **94.45±1.84** |
| A → (B, C, D) | 60.02±3.92 | 75.22±4.41 | 76.08±1.46 | *N/A* | *N/A* | *N/A* | **87.24±2.25** |
| (A, B) → (C, D) | 62.49±2.74 | 81.51±2.47 | 88.09±1.56 | *N/A* | *N/A* | *N/A* | **92.13±1.75** |
| (C, D) → (A, B) | 66.18±3.63 | 80.37±3.00 | 89.01±1.18 | *N/A* | *N/A* | *N/A* | **90.83±1.62** |
| (A, C) → (B, D) | 63.80±2.48 | 79.15±3.85 | 85.21±2.04 | *N/A* | *N/A* | *N/A* | **89.61±1.56** |
| (B, D) → (A, C) | 68.73±2.95 | 81.59±2.71 | 88.88±1.36 | *N/A* | *N/A* | *N/A* | **88.17±1.02** |
| (A, D) → (B, C) | 71.96±2.17 | 83.15±2.76 | 80.69±1.11 | *N/A* | *N/A* | *N/A* | **91.52±1.27** |
| (B, C) → (A, D) | 69.03±1.99 | 82.99±2.73 | 84.28±2.01 | *N/A* | *N/A* | *N/A* | **92.48±1.07** |

# D  Results

Table 3 reports five different types of results for multi-site fMRI analysis. The first section of rows shows the resullts for peer to peer TL models — *i.e.*, applied to datasets where some subjects appear in each pair of sites. The second section of rows analyzes the situation where either the testing set, or

| | Class=Congruent | | Class=Incongruent | |
| :-: | :-: | :-: | :-: | :-: |
| | Dataset: E | Dataset: F | Dataset: E | Dataset: F |

Figure 1: Comparing the site-specific common space in datasets E and F with the global shared space.

the training set, involves two datasets. The third section analyzes peer to peer TL where no subject appears in each pair of sites. This means we cannot use SRM, MDDL, and MDMS as these methods can only transfer cognitive tasks between multi-site fMRI datasets (with overlap) if some subjects appear in each pair of sites. The last two sections repeat experiments on the third section datasets, but using multi-site analysis.

This results reported here are more detailed than in the Experiment section of the main text, which just reports the average each pair of one-way analysis — *e.g.*, $acc(\text{A} \to \text{B})$ and $acc(\text{B} \to \text{A})$ are shown as $acc(\text{A} \rightleftharpoons \text{B})$.

## E   Visualizing transferred neural responses

This section visualizes both the site-specific common space ($\mathbf{G}^{(d,S_d)}$) and the global shared space ($\mathbf{G}^{(d,S_d)}\mathbf{W}$) generated by SSTL for the dataset E and F. We chose these datasets as they include only 2 classes (*i.e.*, Congruent, and Incongruent) — allowing us to visualize all of the stimulus categories. Here, SSTL first generates the site-specific common space in voxel space ($k = V$) for datasets E and F separately. The first row of Figure 1 illustrates the average of these site-specific common space across time points belonging to each category of stimuli — *i.e.*, $\sum_{t \in Y} \mathbf{g}_{t.}^{(d,S_d)}$ for $Y = \{\text{Congruent},$ Incongruent$\}$, and $d = \{\text{E, F}\}$. We then transfer these site-specific common space to global shared space. The second row of Figure 1 shows the average of the transformed common features across time points belonging to each category of stimuli — *i.e.*, $\sum_{t \in Y} \mathbf{q}_{t.}^{(d,S_d)}$ for $\mathbf{Q}^{(d,S_d)} = \mathbf{G}^{(d,S_d)}\mathbf{W}$, $Y = \{\text{Congruent, Incongruent}\}$, and $d = \{\text{E, F}\}$. While the average of these site-specific common features are significantly distinctive for sites E and F, they are highly correlated after transforming to the global shared space.

## F   SRM versus SSTL

The first difference between the earlier SRM [4] versus our SSTL lies in defining the shared space. SRM uses $\min \sum_{s=1}^{S_d} \|\mathbf{X}^{(d,s)} - \mathbf{R}^{(d,s)}\mathbf{G}^{(d,S_d)}\|_F^2$ as the objective function, where $\mathbf{G}^{(d,S_d)} = \sum_{s=1}^{S_d} \left(\mathbf{R}^{(d,s)}\right)^\top \mathbf{X}^{(d,s)}$ and $\left(\mathbf{R}^{(d,s)}\right)^\top \mathbf{R}^{(d,s)} = \mathbf{I}$. Instead, SSTL utilizes $\min \sum_{s=1}^{S_d} \|\mathbf{X}^{(d,s)}\mathbf{R}^{(d,s)} - \mathbf{G}^{(d,S_d)}\|_F^2$ as objective function, where $\left(\mathbf{G}^{(d,S_d)}\right)^\top \mathbf{G}^{(d,S_d)} = \mathbf{I}$. Note that both the site-dependents ($\mathbf{G}^{(d,S_d)}$) and the global shared space ($\mathbf{W}$) in SSTL are orthonormal;

thus, transformation for each site ($\mathbf{W}\,\mathbf{G}^{(d,S_d)}$) is also orthonormal. Like [5], these transformation matrices only apply standard rotations on the neural responses and will preserve the general shape of the data distribution during the transformation procedure. Empirical studies in [6] also showed that the original forms of SRM and HA (*i.e.*, the shared space) can rapidly reduce the performance of multi-site fMRI analysis. Moreover, SRM uses a probabilistic, iterative optimization approach that may coverage to a different $\mathbf{G}^{(d,S_d)}$ in each independent run. We instead propose a single-iteration optimization approach that our empirical studies demonstrate to be more time-efficient and robust.

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

[1]University of Alberta, [2]Nanjing University of Aeronautics and Astronautics, [3]Alberta Machine Intelligence Institute (Amii), [4]Purdue University Fort Wayne

UNIVERSITY OF ALBERTA

amii

easy fMRI

## Motivation

➢ **Functional alignment in a single-site fMRI dataset**

Stimuli · Subjects · An fMRI Machine · Neural Responses X$^{(d,s)}$ · Common Feature Space G$^{(d,Sd)}$

➢ Task-based functional magnetic resonance imaging (fMRI):
  ○ a prevalent tool in neuroscience to analyze how human brains work.
➢ Challenging issues in most fMRI studies:
  ○ **High-dimensionality** and noisy
  ○ **Expensive** to collect with **small sample sizes**
  ○ **Batch effects:** a set of external elements that may affect the performance of analysis

## Shared Space Transfer Learning (SSTL)

➢ The proposed **Shared Space Transfer Learning (SSTL):**
  ○ A novel **Transfer Learning (TL)** approach for **multi-site fMRI analysis**
  ○ It can **functionally align homogeneous** multi-site fMRI datasets
  ○ It **IS NOT LIMITED** to *overlapped datasets* (*i.e.,* share some subjects)
  ○ It can improve the **prediction performance** in every site.
➢ SSTL learns a **TL model** by using a hierarchical **two-step** procedure:
  ○ STEP 1: Extracting a set of **site-specific common features** for each site.
  ○ STEP 2: Transferring the **common features** to a **site-independent, global, shared space.**
➢ SSTL uses a **single-iteration optimization approach**

## SSTL: Objective Functions

➢ **STEP 1: Generating the common space for *each site:***

$$\mathcal{J}_C^{(d)}\left([\mathbf{X}^{(d,s)}]_{s=1\ldots S_d}\right) = \arg\min_{\mathbf{R}^{(d,s)}, \mathbf{G}^{(d,S_d)}} \sum_{s=1}^{S_d} \left\|\mathbf{G}^{(d,S_d)} - \mathbf{X}^{(d,s)}\mathbf{R}^{(d,s)}\right\|_F^2,$$

$$\text{subject to } \left(\mathbf{G}^{(d,S_d)}\right)^{\top}\mathbf{G}^{(d,S_d)} = \mathbf{I}_k.$$

  ○ X$^{(d,s)}$ denotes the **neural responses** for *s-th* subject in *d-th* site
  ○ R$^{(d,s)}$ denotes the **mapping matrices** for *s-th* subject in *d-th* site
  ○ G$^{(d,sd)}$ denotes **the common space** for *d-th* site

➢ **STEP 2: Generating the global shared space**

$$\mathcal{J}_G(\mathbf{G}) = \arg\min_{\mathbf{W}}\left\|\mathbf{G} - \mathbf{GWW}^{\top}\right\|_F^2,$$

$$\text{subject to } \mathbf{W}^{\top}\mathbf{W} = \mathbf{I}_k.$$

  ○ G denotes the concatenated version of **all common spaces in the training set**
  ○ W is the **global shared space**

## Datasets

| ID | Title (Open NEURO ID) | Type | $S_d$ | #1 | $T_d$ | #2 | #3 |
|---|---|---|---|---|---|---|---|
| A | Stop signal with spoken pseudo word naming (DS007) | Decision | 20 | 4 | 149 | B, C | B, C, D |
| B | Stop signal with spoken letter naming (DS007) | Decision | 20 | 4 | 112 | A, C | A, C, D |
| C | Stop signal with manual response (DS007) | Decision | 20 | 4 | 211 | A, B | A, B, D |
| D | Conditional stop signal (DS008) | Decision | 13 | 4 | 317 | | A, B, C |
| E | Simon task (DS101) | Simon | 21 | 2 | 302 | | F |
| F | Flanker task (DS102) | Flanker | 26 | 2 | 292 | | E |
| G | Integration of sweet taste – study 1 (DS229) | Flavour | 15 | 6 | 580 | H | H |
| H | Integration of sweet taste – study 3 (DS231) | Flavour | 9 | 6 | 650 | G | G |

$S_d$ is the number of subject; #1 is the number of stimulus categories; $T_d$ is the number of time points per subjects; #2 lists the other datasets that overlap with this dataset; #3 lists the other datasets whose neural responses can be transferred to this dataset.

## Multi-site classification analysis for pairs of datasets that overlap

A ⇌ B · A ⇌ C · B ⇌ C · G ⇌ H

(A, B) ⇌ C · (A, C) ⇌ B · (B, C) ⇌ A

➢ Multi-site classification analysis for datasets that overlap (*i.e.,* share some subjects). Error bars illustrate ±1 standard deviation.
➢ We compare SSTL with *6 different* existing methods:
  ○ Raw neural responses in **MNI** space without using TL methods
  ○ Shared response model (**SRM**)
  ○ Maximum independence domain adaptation (**MIDA**)
  ○ Side Information Dependence Regularization (**SIDeR**)
  ○ Multi-dataset dictionary learning (**MDDL**)
  ○ Multi-dataset multi-subject (**MDMS**)

## Multi-site classification analysis for sets of datasets that do not overlap

A ⇌ D · B ⇌ D · C ⇌ D · E ⇌ F

(A, B, C) ⇌ D · (A, B, D) ⇌ C · (A, C, D) ⇌ B · (B, C, D) ⇌ A

(A, B) ⇌ (C, D) · (A, C) ⇌ (B, D) · (A, D) ⇌ (B, C)

➢ Multi-site classification analysis for datasets that have no overlap (i.e., do not share any subjects). Error bars illustrate ±1 standard deviation.

## Visualizing transferred neural responses

**Class=Congruent** | **Class=Incongruent**

Dataset: E · Dataset: F · Dataset: E · Dataset: F

Site-specific common space / Global shared space

## Conclusion

In this paper, we propose the *Shared Space Transfer Learning (SSTL)* as a novel transfer learning (TL) technique that can be used for homogeneous multi-site fMRI analysis. Our comprehensive experiments confirmed that SSTL achieves superior performance to other state-of-the-art TL analysis methods. We anticipate that SSTL's multi-view technique for transfer learning will have strong practical applications in neuroscience — such as functional alignment of multi-site fMRI data, perhaps of movie stimuli.