[Reviews · NeurIPS 2020]

Review 1

Summary and Contributions: This paper introduces the Shared Space Transfer Learning (SSTL) as a novel transfer learning technique that can be used for homogeneous multi-site fMRI analysis. SSTL first extracts a set of common features for all subjects in each site. It then uses TL to map these site-specific features to a global shared space, which should improve the performance for classification task.

Strengths: The paper addresses a significant problem: cross-site generalization of machine learning models. The proposed estimator is fairly simple to implement. SSTL uses a scalable optimization procedure that can extract the common features for each site in a single pass through the subjects. Experiments are performed on a reasonable number of datasets for transferring between various cognitive tasks — such as decision making, flavor assessment, etc.

Weaknesses: While the proposed method is really simple, with basic linear algebra operations, it is very hard to follow the presentation, because the objective function is never stated clearly and the method is presented as kind of algorithmic recipe. For instance, regularization terms (rank reduction ridge-like terms) are introduced on the fly with ad hoc justifications, making it hard to understand the whole framework. The cross-site part apparently consists in applying a W matrix of size k*k. I did not understand how this can make datasets closer. Is W site-dependent ? If yes, how do we avoid circularity: using the same data for aligning and predicting... if not, applying W is useless since it is applied before a rotationally invariant classifier (SVM)... but what does the inter-site alignment consist in ? In general, I'm completely missing what is done at test time: this is not made clear in the supplementary material. Equations (6-7) seem to be an awkward way to write stacking across subjects (if I understand correctly). It is unclear whether the final result depends on (arbitrary) subject ordering per site. Why do the authors introduce the bagging over solutions on the training set ?

Correctness: Given all the pending concerns above, it is very hard to assess the correctness of the method: Specifically, I am not sure whether the test data are left untouched during the estimation procedure. How can they be simultaneously aligned and independent from the training data ? I haven't found better explanations in the supplementary materials.

Clarity: The paper is poorly written In equations(2) and (3) the authors introduce a regularization that is not accounted for in Eq (1) Lemma 1 is poorly stated, what does it mean to assume that " some matrices are optimum transformation matrices" l.143 I don't understand; if V>>Td, X.X^T are full rank instead ? l.148 Is equation (5) a definition ? What is supposed ? It is unclear what equation (10) represents: how do theta_1, theta_2 relate to \bar{g} ? What does Omega concretely represent ?

Relation to Prior Work: The paper is a kind of revamping of ref [3] in a multi-site setting. the novelty is not impressive, given that the basic ingredients are re-used. Other works, in particular at NeurIPS, have addressed across-site generalization [Mensch et al. NeurIPS 2017]. I don't find them cited here.

Reproducibility: Yes

Additional Feedback: Note that temporal alignment is not feasible in general: the stimulus order should ideally be randomized across participants, breaking the time consistency of fmRI signals. (https://www.biorxiv.org/content/10.1101/077131v2) "Shared Response Model (SRM) [5] and Hyperalignment [1, 4] are the best examples of multi-view approaches that can align neural responses, but they work most effectively on an individual site." I don't see why ? And as I explained above, I don't understand the technical improvement brought by this paper. Orthogonality constraint in (1) sounds arbitrary and is not justified Fig.1 labels are not readable Experiments are done on smallish datasets (few subjects; how many dimensions ?) Suppmat Fig. 2 what does the colorbar represent ?


Review 2

Summary and Contributions: The paper proposes a transfer learning approach to align fMRI data from multiple sites with subjects performing the same tasks, without requiring subjects to overlap between sites. It achieves this by essentially solving two constrained sum of least squares problems. The first two map all subjects from a single site to a shared space representing that site. The second to map these shared spaces from all sites to a single global space. Both optimizations are performed without a need for an iterative solution. Once these mappings are learnt all subjects' data can be projected to this shared global space and analysis, for example classification can be performed on this.

Strengths: Soundness of claims: The authors claim to propose a method that can utilize fMRI data from multiple sites and improve MVPA since data from more sites can now be utilized to learn a better classification model. In terms of results on classification shown in figure 1, this claim seems to be backed up. The other claim about the optimization procedure not being iterative is also backed up given the derivation. Significance and novelty: Since fMRI data is expensive to gather, any proposal that makes it easier to utilize the existing data available to researchers is definitely significant from an application point of view. Empirical evaluation: providing classification results and runtimes for multiple datasets and comparing with multiple related approaches is a good effort.

Weaknesses: Soundness of claims: In addition to the claims that are indeed backed up. Authors also seem to make claims that might be true but aren't supported/guaranteed in the paper. [see details in Additional feedback] Significance and novelty: While the work is significant from an fMRI application point of view and will help progress neuroscience, the ML contribution of the paper is somewhat limited. Specially given the similarity with SRM. [more in Additional feedback] Empirical evaluation: While the evaluations are fairly extensive both in terms of comparing with other methods and in terms of datasets presented. The reporting of the results seems a tad bit off from the results themselves. [see "Correctness" below]

Correctness: see strengths and weaknesses section.

Clarity: The paper is written well overall. There are some minor typos and minor flow issues [see additional feedback]

Relation to Prior Work: For the most part yes. They could do a better job explaining how their proposal (or at least the first half of it) is different from SRM.

Reproducibility: No

Additional Feedback: First of all, great job in getting a fairly decent work out for review given the circumstances we are in. The paper is well written and the proposal has the makings of a very useful application for neuroscience. I hope my suggestions here will be taken as constructive criticisms, if they sound harsh, my apologies, I am only trying to be as direct as possible. While this might look a like a long list of complaints, I do like the paper and believe that we need these sort of papers that have an obvious impact on progressing neuroscience and bring back some of the "neuro" in Neurips. #1 lines 55 to 57. the claims are that SSTL generates a Robust, Generalized and Accurate MVPA model: First, this sets the reader up to expect guarantees (or at least empirical results) that show robustness, generalizability and accurateness of the the MVPA model trained on data mapped using SSTL. While the accurateness part of it can be established through the presented results, the robustness and generalizability are left to the reader to figure out. It would help if a clear case is made in the experiments section about how the presented results back up the three claims. Second, the statement (and other statements in the paper including the algorithm 1 in supplementary materials and line 194 in main paper) makes it sound like the MVPA model is a part (or a direct product) of SSTL. That doesn't seem to be true. SSTL essentially makes more data available for an MVPA model to be trained (and tested). If there were some sort of feedback loop where the mapping in SSTL are iteratively updated based on classification performance then it'd be appropriate to claim that the resulting MVPA model is a part or product of the method. Right now the improvement in MVPA is from more training data being made available (which of course is a contribution itself). #2 similarity with SRM (equation 1). Eq 1 is the exact same objective function as the shared response model. And given that this first step is performed per site, this step can be performed by SRM. While i understand the the solution presented here does not require iteration as compared to SRM, the connection should be explicitly acknowledged in my view. #3 I have a few questions about the reporting on figure 1 in sections 4.1 and 4.2. First in line 242-243, it says MIDA and SIDer perform better but the don't perform well on multi-site analysis. In figure1 SIDeR does better than at least some of the multi-vew techniques. Similarly, how are all 42 t-tests significant, at least visually in figure 1 (g) MDMS and SSTL basically have the same results. Regarding runtime results, in my view those should be presented in the main manuscript since there is space. Also, it's better to report runtimes in units of time rather than percentage. An improvement of say 5% in runtime is highly subjective on how much time it takes originally. #4 line 41,42: this claim is fairly loaded and should come with some sort of an asterisk. Right now it reads like brain connectomes are deterministic and rigid for each person. If that were true it'd be hard to explain why the same participant's response to the same stimuli may vary across time (or across repetitions). It also makes it sound like having different connectomes is the only reason for differences in neural response between participants. It might be better to tamper this line down a little bit. #5 the links to datasets and toolbox don't work unfortunately. I was excited to try it out. #6 a few typos and writing suggestions. - It might be better to just call MVP analysis as MVPA instead of writing "analysis" separately every time, that is fairly standard acronym in fMRI papers and is well understood. - line 15, can do without the "indeed" - line 19, having two random examples of cog tasks in the abstracts reads strange. the sentence can be closed at tasks. -line 27, MVPA is not in itself a machine learning technique per se, it's just an umbrella term used for any machine learning (generally classification) method applied to multiple voxels. - line 29, MVPA isn't limited to being done out of subject. It could also be used to held out runs for example. - line 37, shared makes it sound like it happened in the past. NIMH and Openneuro are ongoing projects to the best of my understanding. -line 48: is "a" relatively straightforward... -line 72: signal to noise "ratio" -line 72: relies heavily on derived properties is an unclear statement. - line 80: that aim "for" a better accuracy rate (?) - line 86: this is probably MIDA, not MEDA - line 120: ---including , shouldn't this be "i.e." -line 124: do you use whole brain or ROIs in your experiments? -line 130: using 'v' to index 'k' is a little bit confusing, since traditionally 'v' is reserved for voxels. may be use 'k' instead of 'v' and 'K' instead of 'k'. -line 170: common features "have a" minimum distribution ... -line 177: these are parameters, not hyper-parameters i believe. UPDATE: increasing my score to 7 with the understanding that the authors will include all suggestions above.


Review 3

Summary and Contributions: This paper focuses on a valuable topic of handling heterogeneous multi-site fMRI data. The proposed shared space transfer learning method can accurately and efficiently tackle the multi-site fMRI data according to a plenty of comparison experiments.

Strengths: (1) The paper is well organized and the proposed method is well described. (2) This study is closely related to the topics of the NeuriPS conference and contributes to the interdisciplinary areas of neuroscience and computer science. (3) The novelty of the contribution is significant.

Weaknesses: In the manuscript, authors can replace some results figures with tables that helps to provide more space for the visualized results of fMRI in the supplementary materials.

Correctness: Yes. The the claims and methods are correct. The empirical methodology is correct and logical.

Clarity: Yes. The paper is well organized and written. Main body of the this paper clearly describe the proposed method and the experimental results are sufficient to demonstrate the efficiency of their model. Besides, a plenty of visualization and evaluations in the supplementary materials can further support the ideas of this paper.

Relation to Prior Work: Yes. The submission reviews a plenty of previous contributions and clearly describes the noverity of this work. This makes it easier for readers to follow this area and understand the idea of authors.

Reproducibility: Yes

Additional Feedback: According to the feedback of authors, I think they can achieve a better presentation in their camera-ready version. Also, I notice that the authors' feedback to other reviewers focus on the core suggestions and try to improve their study in the right direction.


Review 4

Summary and Contributions: This paper deals with the important problem of small sample size in multi-voxel pattern analysis. A Shared Space Transfer Learning (SSTL) approach is proposed for functionally aligning homogeneous multi-site fMRI datasets. SSTL is compared against a number of state-of-the-art with improved predictive performance shown for a good number of fMRI datasets.

Strengths: SSTL is “multi-view” and does not require some of the subjects to be present in multiple studies unlike other existing multi-view transfer learning methods. A scalable optimization algorithm that works for fMRI datasets with a large number of subjects is proposed. Improved predictive performance is shown over other state-of-the-art methods.

Weaknesses: Minor point, but how is the regularization parameter chosen? Are the results sensitive to this choice? Is homogeneous cognitive tasks necessary for transfer learning? Voxel size of 4mm is very low resolution. For evaluation, are all subjects transferred to common space, and then split into training and test? If so, then data of all subjects have been seen (though not the class labels), and would this introduce correlation between training and test samples hence result in biased accuracy estimates? This problem is analogous to how when we standardize features, we do not use the mean and standard deviation of all subjects but only those of the training subjects. Thus, would be much cleaner to take a portion of the subjects from each site as training, and the rest for testing, which seems to require only a small modification to the proposed method. Is k selected based on validation set? For fair comparison with MNI, feature selection should be performed. In fact, would be interesting to compare against MNI (after feature selection) with global shared space to demonstrate that finding site-specific embedding is necessary.

Correctness: Minor point, but high-dimensional and noise is not the reason why most of the recent neuroimage studies utilize machine learning approaches. Again minor point, but some statements are flawed/contradicting: “Moreover, each person has a different neural response for each individual stimulus because different brains have different connectomes” and “Recent (single site) studies show that the neural responses of all subjects (in that site) can be considered as the noisy rotations of a common template” Need to describe "different" more precisely. If all subjects are completely different, then there won’t be any generalizable pattern to learn.

Clarity: Minor point, but “generalized” MVP analysis does not mean the same thing as generalizable MVP predictor. Generalized in statistics means a model for distributions beyond Gaussian. Need some motivation for site-specific common features beyond aligning to MNI and performing feature selection. Need more context and intuition leading up to equations e.g. (5) and practical procedures. Need more description of notation, e.g. G(d;Sd), but Sd is the number of subjects.

Relation to Prior Work: Not very clear which part of the proposed method is new. From the description, sounds like it is a combination of existing techniques [19-22], hence novel at the approach level.

Reproducibility: Yes

Additional Feedback: I have read the rebuttal, and the authors confirmed that test sites were not used for learning the global shared space, which was an assumption I made when giving the original score. Hence, I can safely keep my original score with this confirmation.

[Author Response · NeurIPS 2020]

We thank all the reviewers for the responses and detailed comments. After reading the feedback, we realized that some
parts of the proposed method might not be explained enough, which might make it difficult to appreciate some of the
motivations and novelties of this paper. We hope that this response may clarify any misunderstanding, and we will
revise the article accordingly.

**SRM versus SSTL:** The first difference between the earlier SRM versus our SSTL lies in defining the shared space.
SRM uses $\min \sum_{s=1}^{S_d} \|\mathbf{X}^{(d,s)} - \mathbf{R}^{(d,s)}\mathbf{G}^{(d,S_d)}\|$ as the objective function, where $\mathbf{G}^{(d,S_d)} = \sum_{s=1}^{S_d} \left(\mathbf{R}^{(d,s)}\right)^{\top} \mathbf{X}^{(d,s)}$
and $\left(\mathbf{R}^{(d,s)}\right)^{\top} \mathbf{R}^{(d,s)} = \mathbf{I}$. Instead, SSTL utilizes $\min \sum_{s=1}^{S_d} \|\mathbf{X}^{(d,s)}\mathbf{R}^{(d,s)} - \mathbf{G}^{(d,S_d)}\|$ as objective function, where
$\left(\mathbf{G}^{(d,S_d)}\right)^{\top}\mathbf{G}^{(d,S_d)} = \mathbf{I}$. Note that both the site-dependents ($\mathbf{G}^{(d,S_d)}$) and the global shared space ($\mathbf{W}$) in SSTL are
orthonormal; thus, transformation for each site ($\mathbf{W}\,\mathbf{G}^{(d,S_d)}$) is also orthonormal. Like [1], these transformation matrices
only apply standard rotations on the neural responses and will preserve the general shape of the data distribution during
the transformation procedure. Empirical studies in [3] also showed that the original forms of SRM and HA (*i.e.*, the
shared space) can rapidly reduce the performance of multi-site fMRI analysis. Moreover, SRM uses a probabilistic,
iterative optimization approach that may coverage to a different $\mathbf{G}^{(d,S_d)}$ in each independent run. We instead propose a
single-iteration optimization approach that our empirical studies demonstrate to be more time-efficient and robust.

**Stacking across subjects in equations (6-7):** Yes, this subject ordering can matter, but this is fairly standard — *i.e.*,
this is true for incremental PCA, SRM, or any stochastic algorithm. Further, our empirical results, over many datasets,
demonstrate that this approach works effectively.

**Learning procedure:** We used a scheme similar to the one proposed in [2–3] for evaluating all transfer learning
approaches described in this paper. In SSTL, we first compute the unsupervised site-dependent $\mathbf{G}^{(d,S_d)}$, from the data,
BUT NOT THE LABELS, for all sites. Note this is similar to the procedures used in learning any classical functional
alignment, such as SRM and HA. For classifying a subject in site $d$, we then use the labeled data from other $d-1$ sites
to find the global shared space $\mathbf{W}$, then train the classifier — *n.b.*, using nothing from the $d$-th site. Hence, we never
use any labels from the $d$-*th* site, when computing the labels for those $d$-*th* site subjects. Like Westfall *et al.* 2017,
we also used the standard learning procedure, *i.e.*, using a shuffled form of neural responses for training the classifier
(not the temporally aligned version). This is currently in Sec 4 (lines 213-223); the actual algorithm appears in the
Supplementary Material. The revised version will explicitly summarize the entire training and performance processes.

**Reviewer 1:** Thank you for your insightful comments.

1) SSTL uses a two-step procedure for analyzing multi-site fMRI datasets. The primary objective functions are based
on equations (4) and (11) on pages 3 and 5. To define (4), we propose (1) as the appropriate form for generating the
site-dependent feature space $\mathbf{G}^{(d,S_d)}$. We learn this using the regularized projection matrix in (2–3). Lemma 1 proves
that we can calculate a regularized version of $\mathbf{G}^{(d,S_d)}$ by substituting (2) in (1). Note that (1–3) are all involved in
Lemma 1's (4). In other words, the regularization is defined for (2), not (1), and appears in (4) after we use (2) to
estimate the site-dependent in (1). We will re-structure Sec 3.1, to better show this flow.

2) We NEVER said that *if $V \gg T_d$, then the scatter matrices are full rank*. Instead, we said that 'scatter matrices
$\mathbf{X}^{(s,d)}\left(\mathbf{X}^{(s,d)}\right)^{\top}$ will be singular and non-invertible', which means these matrices are NOT full rank. The papers [3–5,
19] explains that $V \gg T_d$ implies the singular scatter matrices.

3) Lines 172 and 174 show how $\bar{\mathbf{g}}_t$ use $\theta_2$ , and $\mathbf{q}_t$, which in turn uses $\theta_1$.

4) Thanks for the references; the revised manuscript will cite those papers — *i.e.*, Mensch 2017 and Westfall 2017.

**Reviewer 2:** Thanks for the very useful feedback! The revised version will address all of your suggestions in the
"Additional feedback".

**Reviewer 3:** Thanks for the great comments! We will provide a better structure for presenting the proposed method
and the results in the camera-ready.

**Reviewer 4:** Thanks for the wonderful comments. The revision will address all of those comments. Indeed, SSTL
can be applied to multi-site fMRI datasets with any resolution. Further, we will provide the MNI results after feature
selection and address the Clarity and Weakness sections' questions.

Thanks to all of your insightful comments, our paper now better shows that SSTL provides an effective way to analyze
multi-site fMRI data. We anticipate that this approach can be used in various mental health applications, and will
contribute to techniques that can help save people's lives. We hope that the reviewers and Area Chair agree, and will
also support publishing this paper.

[Meta-Review · NeurIPS 2020]

The reviewers found that this paper was useful for the field, offering a new method for aligning multi-site fMRI data, with some disagreement. One of the main concerns was about the clarity of the paper which greatly impacts its usefulness. Please improve the quality of the writing for the final draft. There is a major typo in the paper: everywhere XX^T is mentioned, I think you mean X^TX (the identify matrices are also wrongly subscripted). In fact R1's comment is right, XX^T is likely not singular given V>>T.